# Linking Motor and Cognitive Decline in Aging: Gait Variability and Working Memory as Early Markers of Frailty

**DOI:** 10.3390/healthcare13243201

**Published:** 2025-12-07

**Authors:** Elisa Valeriano-Paños, Mª Nieves Moro-Tejedor, Mª Jesús Santamaria-Martin, Susana Vega-Albala, María Valeriano-Paños, Juan Francisco Velarde-García, Luis Enrique Roche-Seruendo

**Affiliations:** 1Faculty of Nursing, Physiotherapy and Podology, University Complutense of Madrid, 28040 Madrid, Spain; elisa.valeriano@salud.madrid.org; 2Department of Nursing, Hospital General Universitario Gregorio Marañón, 28007 Madrid, Spain; mnieves.moro@salud.madrid.org; 3Research Nursing Group of Instituto de Investigación Sanitaria Gregorio Marañón (IiSGM), 28007 Madrid, Spain; 4Los Yébenes Health Service, Madrid Health Service, 28047 Madrid, Spain; mariajesus.santamaria@salud.madrid.org; 5Peña Prieta, Health Service, Madrid Health Service, 28038 Madrid, Spain; susana.vega@salud.madrid.org; 6Faculty of Psychology and Speech Therapy, Valencia University, 46010 Valencia, Spain; maria.valeriano@ovalopsicologia.com; 7Research Group in Social Health Care Needs for the Population at Risk of Exclusion, School of Nursing, Red Cross University, Autonomous University of Madrid, 28003 Madrid, Spain; 8Research Group of Humanities and Qualitative Research in Health Science (Hum & QRinHS), Universidad Rey Juan Carlos, 28922 Alcorcón, Spain; 9Department of Physical Therapy, San Jorge University, 50830 Zaragoza, Spain; leroche@usj.es

**Keywords:** frailty, gait variability, working memory, executive function, older adults, mild cognitive impairment

## Abstract

**Highlights:**

**What are the main findings?**
Shorter stride length during fast walking, mild cognitive impairment, depressive symptoms, and female sex were identified as significant predictors of the transition from non-frail to prefrail status.Additionally, increased stride time variability at fast pace and poorer working memory performance were independently associated with the progression from prefrailty to frailty.

**What are the implications of the main findings?**
Spatiotemporal gait variability and executive dysfunction represent sensitive multidomain markers for the early detection of frailty in community-dwelling older adults.Integrating gait and cognitive assessments into routine geriatric evaluations may enhance early identification and prevention efforts, supporting a multidimensional approach to aging care.

**Abstract:**

**Background/Objectives:** Frailty is an age-related clinical syndrome characterized by diminished physiological reserves and increased vulnerability to adverse outcomes. Growing evidence suggests that frailty involves shared brain networks that regulate both gait and cognitive functions. This study aimed to examine the relationship between frailty status, spatiotemporal gait parameters, and cognitive functions in community-dwelling older adults. **Methods:** A cross-sectional study was conducted with 99 adults aged ≥70 years, classified as non-frail, prefrail, or frail according to the Fried phenotype. Gait parameters were measured under usual and fast walking conditions using the OptoGait^®^ photoelectric system. Cognitive status was assessed with the Montreal Cognitive Assessment (MoCA) and a comprehensive neuropsychological battery. Multivariate logistic regression analyses were performed to identify factors associated with transitions between frailty stages. **Results:** The prevalence of frailty was 9.1%, with 51.5% prefrail and 39.4% non-frail. The transition from non-frail to prefrail was associated with shorter stride length at fast pace (OR = 0.92, 95% CI: 0.88–0.96), mild cognitive impairment (OR = 3.71, 95% CI: 1.08–12.69), depressive symptoms (OR = 1.82, 95% CI: 1.26–2.62), and female sex (OR = 4.94, 95% CI: 1.20–16.77). The transition from prefrail to frail was linked to increased stride time variability at fast pace (OR = 2.94, 95% CI: 1.34–6.44) and poorer working memory (OR = 0.40, 95% CI: 0.16–0.97)**. Conclusions:** Shorter stride length, mild cognitive impairment, and depressive symptoms emerged as key markers of the transition from non-frailty to prefrailty, whereas increased stride time variability and poorer working memory distinguished prefrail from frail individuals. These findings highlight gait- and executive-function-related markers as sensitive early indicators of vulnerability. Incorporating quantitative gait assessment and brief cognitive screening into routine geriatric evaluations may substantially enhance early detection and support targeted preventive strategies for healthy aging.

## 1. Introduction

Frailty is a complex clinical condition associated with aging, with a significant impact on the health and quality of life of older adults [1]. Within the context of global population aging, it has become an increasingly relevant public health concern, with prevalence rates reaching approximately 9% for physical frailty and 18.8% for cognitive and social frailty [2,3]. These estimates, however, may vary depending on the assessment tool used and the characteristics of the population under study [4]. Among the available instruments, the frailty phenotype proposed by Fried et al. [5] is the most widely accepted and frequently cited in the scientific literature [6]. The rapid demographic expansion of older populations worldwide suggests that the number of individuals living with frailty will continue to rise, consolidating frailty as a major public health challenge that requires targeted strategies for early detection and prevention [7].

From a clinical and pathophysiological perspective, frailty is characterized by diminished physiological reserves and an impaired ability to maintain homeostasis, distinguishing it from comorbidity or disability [8]. Contemporary models conceptualize it as a dynamic, multidimensional process resulting from the cumulative dysregulation of multiple biological systems rather than a single disease [9]. Key contributing pathways include chronic low-grade inflammation, vascular dysfunction, endocrine alterations, sarcopenia, and neurodegeneration [10,11]. Some of these mechanisms—particularly vascular pathology and systemic inflammation—may act simultaneously as antecedents that increase vulnerability and as consequences that worsen once frailty is established, creating bidirectional relationships with cardiovascular disease and cognitive impairment [12,13].

There is broad consensus that frailty increases susceptibility to adverse outcomes [14]. When exposed to stressors such as acute infection or environmental challenges, frail individuals are at higher risk of experiencing rapid functional decline, hospitalization, institutionalization, falls, or even premature death compared with robust older adults [8,15]. Physical inactivity is one of the major risk factors for frailty, as sedentary behavior is associated with insulin resistance, elevated inflammatory markers, prothrombotic states, and increased cardiovascular risk—all of which have been strongly linked to frailty development [16].

More than a decade ago, researchers proposed that the physiological mechanisms underlying frailty might also be implicated in brain aging and cognitive decline [17]. Since then, epidemiological and clinical studies have consistently demonstrated the association between frailty and cognitive impairment [18], giving rise to the concept of cognitive frailty. This construct expands the classical definition of frailty by incorporating a reduction in cognitive reserve—typically reversible—alongside physical decline, often reflected in reduced gait speed [19]. This conceptualization aligns with emerging theoretical models suggesting that early gait disturbances and cognitive deficits may reflect the vulnerability of shared neural networks affected by modifiable factors such as vascular damage and chronic inflammation [20].

At the same time, the progressive aging of urban populations presents new challenges for the early identification of both frailty and cognitive decline [21,22]. Current evidence indicates that subtle alterations in gait and executive function may emerge even before major clinical syndromes such as dementia become apparent, making them potential early markers of vulnerability [23,24,25,26,27]. However, the precise characterization of these changes across different stages of frailty remains limited [28,29].

Recent systematic reviews and longitudinal analyses have sought early markers capable of predicting frailty before the syndrome becomes clinically evident. Subtle gait disturbances, particularly reduced stride length, slower gait speed, and increased variability, have been identified as early motor indicators of emerging frailty and future adverse outcomes [30,31,32,33,34]. Similarly, cognitive changes in executive function, attention, and working memory have been shown to precede the transition from robustness to prefrailty, reinforcing the notion of cognitive frailty as an early expression of vulnerability in shared brain networks [17,18,23,27,35]. Moreover, concurrent declines in gait and cognition have been identified as strong predictors of incident frailty, cognitive impairment, and disability among community-dwelling older adults [23,24,25]. Together, these findings suggest that motor and cognitive markers—individually and in combination—may serve as sensitive early predictors of frailty, underlining the need to investigate their role in distinguishing between frailty stages.

To guide the present study, the following research questions were formulated: Q1: Are spatiotemporal gait parameters—particularly stride length and stride time variability—associated with different frailty stages in community-dwelling older adults? Q2: Do cognitive functions, especially the presence of mild cognitive impairment and working memory performance, help differentiate transitions between frailty stages?

Accordingly, the aim of this study was to examine the relationship between frailty, spatiotemporal gait parameters, and cognitive functions in non-institutionalized older adults living in an urban setting. We hypothesized that frail individuals aged 70 years or older without dementia would exhibit greater stride time variability and poorer executive function performance.

## 2. Materials and Methods

### 2.1. Design

A cross-sectional observational study was conducted to examine the relationship between frailty, spatiotemporal gait parameters, and cognitive function in older adults. This study was conducted in accordance with the Strengthening the Reporting of Observational Studies in Epidemiology (STROBE) guidelines [36].

### 2.2. Participants

Participants were recruited from two primary care centers of the Madrid Health Service in Spain (SERMAS). The inclusion criteria were: being 70 years of age or older, scoring ≥24 on the MEC-30 (Mini-Examen Cognoscitivo, the Spanish version of the Mini-Mental State Examination), being able to walk 100 m unassisted, speaking Spanish fluently, and living independently in the community. The minimum age of 70 years was selected because frailty prevalence and prefrailty rates increase markedly from this age onward, and early motor and cognitive changes become more clinically detectable. Moreover, this threshold aligns with standardized frailty screening practices within the Spanish primary care system, where structured assessments typically begin at age 70. Exclusion criteria included institutionalization, lack of comprehension of the Spanish language, and inability to complete the frailty or cognitive assessments.

### 2.3. Sample Size

The required sample size was estimated using the standard formula for calculating a population proportion with a specified precision. Because the primary objective was to estimate and compare frailty categories in a cross-sectional design, the calculation was based on the expected prevalence of frailty rather than on an effect-size–based model. We used an expected frailty prevalence of 14% in community-dwelling older adults, derived from epidemiological data in Spanish primary care populations [37]. Using a 95% confidence level and a margin of error of 7%. This approach is widely recommended for prevalence studies in which the primary parameter of interest is the proportion of individuals within a given frailty category.

### 2.4. Data Collection

Data were collected individually at the participating healthcare centers during two scheduled visits approximately one week apart. During the first visit, sociodemographic, anthropometric, clinical, and physical function data were obtained through standardized interviews and validated measurement procedures. The second visit was dedicated to the administration of cognitive assessments by trained evaluators under controlled conditions.

All assessments were conducted by qualified healthcare professionals who received specific training to ensure inter-rater reliability. The same instruments and testing protocols were applied across both centers to maintain methodological consistency.

Validated and standardized instruments were used to ensure accuracy and reproducibility of all measures. The variables assessed provided a comprehensive profile of each participant’s health status, physical performance, frailty level, gait characteristics, and cognitive function.

### 2.5. Measures and Instruments

Frailty was assessed according to the phenotype model proposed by Fried et al., with specific adaptations for this study [8]. This approach remains the most widely validated and commonly applied framework for identifying physical frailty in community-dwelling older adults [8]. Frailty was operationalized using the five original criteria described by Fried et al.: (1) Unintentional weight loss, defined as a self-reported loss of ≥4.5 kg or ≥5% of body weight over the previous year; (2) Exhaustion, measured using two items from the CES-D scale (“I felt that everything I did was an effort” and “I could not get going”), with a response of “frequently” (≥3–4 days/week) indicating exhaustion; (3) Slowness, evaluated through gait speed assessed with the OptoGait^®^ photoelectric system under usual walking conditions—participants with a mean walking speed below 1 m/s were classified as meeting the slowness criterion, consistent with recent recommendations for frailty screening [28]; (4) Weakness, assessed using a T.K.K. 5101 GRIP-D handgrip dynamometer (Takei Scientific Instruments, Japan), applying sex- and body mass index (BMI)-specific cut-off points; and (5) Low physical activity, measured with the Physical Activity Scale for the Elderly (PASE), with scores below the sex-adjusted 20th percentile indicating low activity levels. Based on the number of criteria met, participants were classified as frail (≥3 criteria), prefrail (1–2 criteria), or non-frail (0 criteria).

Quantitative gait analysis was conducted using the OptoGait^®^ photoelectric system, a validated and reliable tool when compared with the GAITRite^®^ system [38]. Participants completed eight walking trials at both usual and fast paces along a 7 m instrumented walkway. Spatiotemporal gait parameters included gait speed, stride length, stride time, support time, swing time, and double support time. Coefficients of variation (CV = standard deviation/mean × 100) were also calculated to assess step-to-step variability. Recent evidence highlights these gait metrics—particularly stride time variability and stride length—as sensitive indicators of early frailty and cognitive decline [39,40].

Global cognitive function was assessed using the Montreal Cognitive Assessment (MoCA), a widely validated tool for detecting mild cognitive impairment (MCI) in older adults. The MoCA evaluates multiple cognitive domains, including visuospatial and executive functions, naming, attention, language, abstraction, delayed recall, and orientation, with a total possible score of 30 points. The test was administered individually in a quiet room following standardized procedures by trained evaluators. To ensure accurate interpretation, the Spanish version of the MoCA was used, applying age- and education-adjusted normative data recently established for Spanish-speaking populations [41,42]. These adjustments account for known sociocultural and educational differences that affect cognitive screening outcomes in Spain and Latin America. MCI was defined as performance below the corresponding adjusted cut-off values.

In addition, specific cognitive domains were assessed using a battery of validated neuropsychological tests. Because working memory, executive function, and attention are related yet distinct cognitive domains, they were assessed with separate neuropsychological measures to avoid conceptual overlap. MoCA domain subscores were used for descriptive purposes only and were not interpreted as standalone measures of executive or attentional functioning. Executive function was evaluated using the Trail Making Test B (cognitive flexibility) and the Stroop test (inhibitory control). Working memory was assessed with the Digit Span Backward test, which is widely recognized as an independent measure of this domain. Divided attention was evaluated using the Digit Symbol-Coding test, and sustained attention was assessed through the Trail Making Test A. Memory was evaluated through immediate recall using the Digit Span Forward test and long-term recall using the delayed free recall trial of the Spain-Complutense Verbal Learning Test [43].

### 2.6. Statistical Analysis

Quantitative variables were summarized as means and standard deviations (SD) when normally distributed, or as medians with interquartile ranges (IQR) when the distributions were non-normal. Categorical variables were presented as absolute and relative frequencies (n, %). The normality of continuous data was assessed to determine the appropriate statistical tests.

Associations between frailty status (non-frail, prefrail, frail) and categorical variables were analyzed using chi-square tests, while continuous variables were compared using one-way analysis of variance (ANOVA) or Kruskal–Wallis tests, depending on data distribution. Assumptions of normality and homoscedasticity were checked prior to conducting ANOVA. In all one-way ANOVA models, frailty status (non-frail, prefrail, frail) was treated as the independent variable. The dependent variables were the continuous clinical, gait, and cognitive measures evaluated in the study, including gait speed, stride length, stride time, coefficients of variation, total MoCA score, and performance on the neuropsychological tests (TMT-A, TMT-B, Stroop interference score, Digit Span Forward and Backward, and Digit Symbol-Coding).

Variables that showed statistically significant associations with frailty (*p* ≤ 0.05) were entered into multivariate binary logistic regression models to identify independent predictors of frailty status. A backward stepwise elimination method was applied, retaining only those variables with *p*-values ≤ 0.05 in the final model. Two separate regression models were constructed: one comparing non-frail and prefrail individuals, and another comparing prefrail and frail individuals.

All statistical analyses were conducted using Stata versión 16^®^. A two-sided *p*-value ≤ 0.05 was considered statistically significant.

### 2.7. Ethical Considerations

The study was conducted in accordance with the ethical principles outlined in the Declaration of Helsinki [44] and the recommendations of the International Committee of Medical Journal Editors (ICMJE) [45]. The study was approved by an appropriate ethics committee. All participants provided written informed consent prior to their inclusion. Participant confidentiality and data anonymity were rigorously maintained throughout the study.

## 3. Results

A total of 99 individuals participated in the study. The mean age of the sample was 78.1 ± 5.1 years, and 58.6% (n = 58) were women. The prevalence of frailty was 9.1% (n = 9), prefrailty 51.5% (n = 51), and non-frailty 39.4% (n = 39). Among prefrail individuals, the most common Fried criteria were slowness (60.8%) and exhaustion (29.4%). In frail individuals, low physical activity and slowness were present in 100% of cases. Age was significantly associated with frailty status (*p* = 0.027). The frail group also showed the highest proportion of individuals living alone, as well as the lowest levels of education and monthly income (Table 1).

Analysis of clinical and epidemiological variables showed that the frail group had the highest comorbidity index compared with the non-frail and prefrail groups. Hyperlipidemia (64.7% and 88.9% vs. 56.4%), diabetes mellitus (35.3% and 55.6% vs. 23.1%), and heart failure (15.7% and 22.2% vs. 15.4%) were more prevalent among the prefrail and frail groups than among the non-frail group. Higher levels of depressive symptoms were observed in groups with more advanced frailty status, with significant differences across all three groups (*p* < 0.001), and specifically between the non-frail and prefrail groups (*p* = 0.001). The number of medications taken daily was also higher in the frail group. Regarding health status, no significant differences were observed in body mass index (BMI) or the Charlson Comorbidity Index (CCI) across the frailty groups. However, depressive symptoms were significantly higher among individuals classified as prefrail and frail compared with non-frail participants. The frail group had the highest scores on the Geriatric Depression Scale (GDS-15), with significant differences observed among non-frail, prefrail, and frail individuals (*p* < 0.001).

Regarding functional ability, Snellen test scores were significantly lower between the non-frail and prefrail groups (*p* = 0.020). Additionally, poorer left-eye visual acuity was significantly associated with frailty status when comparing all three groups (*p* = 0.047). Dependence in Activities of Daily Living (ADL) was strongly related to frailty (*p* < 0.001), as was the need for assistance when climbing and descending stairs (*p* < 0.001). Several gait parameters at both usual and fast walking speeds were significantly associated with frailty (Table 2).

At fast pace, the coefficient of variation (CV) of stride time differed significantly between the non-frail and frail groups (*p* = 0.008) and between the prefrail and frail groups (*p* = 0.042), but not between non-frail and prefrail individuals. In terms of global cognitive function (Table 3), MoCA performance declined progressively with increasing frailty, with significant differences across the three groups and specifically between the non-frail and prefrail groups (*p* = 0.021). Within the MoCA domains, attention was the only cognitive domain significantly associated with frailty status across all three groups (*p* = 0.019) and between non-frail and prefrail participants (*p* = 0.004). The prevalence of mild cognitive impairment (MCI) was higher in prefrail (47.1%) and frail individuals (44.4%) compared with non-frail individuals (20.5%), with statistically significant differences both across all groups (*p* = 0.030) and between the non-frail and prefrail groups (*p* = 0.008). As recommended in cognitive assessment guidelines, MoCA subscores were interpreted solely as supportive indicators and not as stand-alone measures of executive or attentional functioning. Regarding specific cognitive functions (Table 3), statistically significant differences were observed in the TMT-B (*p* = 0.040) and DSC (*p* = 0.042) tests between the non-frail and frail groups, with poorer performance in frail participants. For the DSB test, significant differences (*p* = 0.037) were found between the prefrail and frail groups, again with worse performance in the frail group.

Factors associated with the comparison between the non-frail and prefrail groups included: (1) female sex (OR = 4.94, 95% CI [1.20–16.77], *p* = 0.025); (2) presence of depressive symptoms (OR = 1.82, 95% CI [1.26–2.62], *p* = 0.001); (3) mild cognitive impairment (OR = 3.71, 95% CI [1.08–12.69], *p* = 0.037), and (4) shorter stride length at fast pace (OR = 0.92, 95% CI [0.88–0.96], *p* < 0.001) (Table 4). In contrast, the transition from prefrail to frail status was significantly associated with: (1) higher stride time variability at fast pace (OR = 2.94, 95% CI [1.34–6.44], *p* = 0.007), and (2) impaired working memory (OR = 0.40, 95% CI [0.16–0.97], *p* = 0.050).

## 4. Discussion

The aim of this study was to examine the relationship between frailty, spatiotemporal gait parameters, and cognitive function in non-institutionalized older adults living in an urban setting. The main findings suggest a progressive association between specific gait and cognitive parameters and the different stages of frailty. Specifically, shorter stride length at fast walking pace was significantly associated with prefrailty, while increased stride time variability at fast pace was linked to frailty. Mild cognitive impairment (MCI), as assessed by the MoCA, was significantly associated with prefrailty, whereas poorer working memory (as measured by the Digit Span Backward test) was significantly associated with frailty.

There is ongoing debate in the literature regarding the role of gait characteristics within the frailty continuum. Gait speed has been the most commonly studied parameter, with values below 0.8 m/s typically associated with frailty and speeds between 0.95 and 1.05 m/s observed in prefrail individuals [30]. However, gait speed alone may lack specificity, as it can be influenced by a range of factors unrelated to frailty. In contrast, our study found that stride length at fast pace was a more sensitive indicator of prefrailty, consistent with prior research showing that stride length and double support time are among the most discriminative parameters for identifying frailty stages [38]. Reduced stride length has also been associated with adverse outcomes, such as falls, disability, and diminished physiological reserve [32], particularly under fast walking conditions, which demand greater motor and cognitive resources [33].

Stride time variability at fast pace was the gait parameter most strongly associated with frailty. These findings align with previous studies indicating that increased variability in stride timing reflects impairments in automatic gait control and is a marker of gait instability and fall risk [34]. Importantly, stride time variability appears to be associated with higher-order cognitive processes, particularly executive control, rather than musculoskeletal limitations [46,47,48].

Cognitive performance was lower in groups with more advanced frailty status. Lower MoCA scores and a higher prevalence of MCI were observed in both prefrail and frail participants, supporting previous findings that cognitive decline is closely intertwined with physical frailty. As reported by Lorenzo-López et al., cognitive performance tends to decline progressively across the continuum from robustness to frailty, indicating that even the prefrail stage is associated with detectable cognitive impairment [35].

Higher-order cognitive domains, including executive control and attention, have emerged as key areas implicated in frailty [18], and our findings reinforce this association. Specifically, poorer working memory performance—considered a distinct yet related cognitive domain—was independently associated with frailty, consistent with prior studies linking working memory deficits to prefrontal dysfunction in frail older adults [49]. Although other executive and attention measures (e.g., TMT-A/B, DSC) showed non-significant trends, these patterns suggest the presence of early, subclinical cognitive changes in the prefrail stage that may not yet be fully detectable using standard neuropsychological tests.

MCI emerged as a distinguishing cognitive feature of the prefrail group, supporting recent evidence that cognitive changes may arise before the onset of full frailty. These findings align with the expanding concept of “cognitive prefrailty,” defined as the coexistence of early physical vulnerability and subtle cognitive deficits in the absence of dementia [50]. Large-scale epidemiological data also indicate that the highest prevalence of MCI occurs among prefrail older adults, reinforcing the notion of this intermediate and potentially reversible stage [35].

The lack of significant differences in specific cognitive domains between non-frail and prefrail individuals may reflect the preserved cognitive reserve typically seen in the prefrail stage, during which compensatory mechanisms may still operate effectively [35]. Another plausible explanation lies in the limited sensitivity of currently available screening tools to detect early, subclinical changes in executive and attentional domains [51]. Overall, these findings align with contemporary evidence suggesting that MCI represents a transitional cognitive state within the frailty continuum, marking an optimal window for early detection and intervention.

Depressive symptoms were more pronounced in participants with higher frailty levels. This finding is consistent with previous evidence highlighting the strong bidirectional relationship between frailty and depression [52]. Shared psychosocial and biological risk factors—such as limited personal and social resources, low educational attainment, income insecurity, and vascular pathology—may contribute to the co-occurrence and mutual reinforcement of both conditions [53,54].

Recent evidence indicates that women are more likely to develop frailty than men, due to biological and social factors such as lower muscle mass, hormonal differences, and greater life expectancy. These sex differences influence both the onset and progression of frailty throughout aging [55]. In our sample, the most prevalent frailty criteria among women were gait slowness and exhaustion, consistent with previous reports highlighting greater psychological vulnerability in women, which may contribute to frailty development [56,57].

Although underexplored in the literature, visual impairment also showed a significant association with frailty, particularly with prefrailty status. Our findings are consistent with prior studies [58], and the observed association between poor left-eye visual acuity and frailty in our sample raises the hypothesis of right-hemisphere vulnerability to vascular or inflammatory damage—a potential avenue for future research.

### 4.1. Limitations

This study has several limitations. As a cross-sectional design, it does not allow for the establishment of causal relationships or temporal sequences. Longitudinal studies are needed to further explore the dynamic interplay between gait, cognition, and frailty progression. Additionally, the use of convenience sampling limits the generalizability of the findings to broader populations. The relatively high baseline functioning of participants may also lead to an underestimation of associations, suggesting a conservative bias. The neuropsychological battery, while validated, may not fully capture early or subtle cognitive changes. Future studies should consider incorporating more sensitive cognitive assessments and neuroimaging techniques.

A major strength of this study is the use of a low-cost, clinically applicable, and reliable electronic system (OptoGait^®^) for objective gait analysis in a real-world setting. This strengthens the feasibility of implementing similar assessments in primary care and geriatric services.

### 4.2. Relevance for Clinical Practice

Early identification of frailty is essential for preventing functional decline and guiding targeted interventions in older adults. The present findings provide several clinically actionable markers that can be integrated into routine assessment pathways in primary care and geriatric practice.

First, shorter stride length at fast walking pace emerged as an early indicator of prefrailty. Because gait speed alone can miss subtle gait disturbances, incorporating fast-paced gait assessment into routine evaluations may improve early detection of vulnerability. Portable tools such as OptoGait^®^ or simple timed walkway tests can be feasibly implemented in primary care to quantify gait performance with greater precision.

Second, increased stride time variability at fast pace was a key discriminator of frailty, reflecting reduced gait automaticity and impaired motor control. This parameter may help identify older adults at higher risk of falls, disability, and functional decline. Clinicians can consider abnormal gait variability as a trigger for referral to physiotherapy, structured balance and strength programs, or fall-prevention interventions.

Third, the association between working memory deficits and frailty, together with the higher prevalence of mild cognitive impairment, underscores the need to incorporate basic cognitive screening—such as the MoCA or brief executive function tests—into frailty evaluations. Detecting cognitive vulnerability enables clinicians to design more tailored interventions, as cognitive impairment can influence adherence to exercise programs, dual-task mobility, and overall self-management.

Fourth, the strong association between depressive symptoms and prefrailty highlights the importance of routinely assessing mood when evaluating frailty risk. Early detection and treatment of depressive symptoms may help mitigate functional decline by addressing modifiable psychological and behavioral factors that contribute to reduced activity and poorer mobility.

Finally, the higher likelihood of frailty among women emphasizes the need for a sex-sensitive approach in frailty screening, acknowledging the greater burden of exhaustion, slower gait, and psychosocial vulnerability often observed in female patients.

In addition to these clinical markers, several evidence-based strategies have proven effective in preventing or slowing functional decline. Multicomponent exercise programs—including resistance training, balance exercises, and fast-paced gait practice—are among the most robust interventions for improving mobility, muscle strength, and gait stability, particularly in prefrail individuals. Cognitive stimulation and dual-task training may further support functional independence by enhancing executive processes involved in gait regulation. Early management of vision problems, depressive symptoms, and polypharmacy also plays a critical role in reducing vulnerability to adverse outcomes and preserving functional capacity. Overall, integrating gait analysis, cognitive screening, and emotional and sensory health assessments into standard frailty evaluations may enhance early detection, support individualized care planning, and strengthen prevention strategies aimed at promoting healthier aging trajectories.

## 5. Conclusions

This study demonstrates that specific gait and cognitive markers play a critical role in differentiating stages of frailty among community-dwelling older adults. Shorter stride length at fast walking pace, the presence of mild cognitive impairment, depressive symptoms, and female sex were independently associated with the transition from non-frailty to prefrailty, highlighting early multidomain changes that signal emerging vulnerability. The contrast between prefrail and frail groups was characterized by increased stride time variability under fast walking conditions and poorer working memory performance, indicating the combined deterioration of motor control and executive function.

These findings reinforce the view that frailty is a dynamic and multidimensional process and that subtle impairments in gait and cognition can serve as sensitive early markers of risk. Incorporating quantitative gait analysis and brief cognitive assessments into routine geriatric and primary care evaluations may enhance early detection, enable timely intervention, and ultimately support healthier aging trajectories.

## Figures and Tables

**Table 1 healthcare-13-03201-t001:** Sociodemographic Characteristics and Their Association with Frailty Syndrome.

Measure	Total (n = 99)	Not Frail (n = 39)	Prefrail (n = 51)	Frail (n = 9)	*p*-Value
Sex n (%)					
Males	41 (41.4%)	19 (48.7%)	21 (41.2%)	1 (11.1%)	0.119
Females	58 (58.6%)	20 (51.3%)	30 (58.8%)	8 (88.9%)	
Cohabitation n (%)					
Alone	29 (29.3%)	14 (35.9%)	11 (21.6%)	4 (44.4%)	0.329
Spouse	65(65.7%)	24 (61.5%)	37 (72.5%)	4 (44.4%)	
Child/other relative	5 (5.1%)	1 (2.6%)	3 (5.9%)	1 (11.1%)	
Age mean ±	78.1 ± 5.1	76.6 ± 4.4	78.5 ± 5.3	81.7 ± 5.4	0.027
Education (years) mean ±	8.0 ± 3.4	8.0 ± 3.0	8.3 ± 3.4	6.7 ± 4.6	0.453
Monthly income € mean ±	1602.5 ± 796.9	1627.4 ± 817.7	1609.1 ± 791.3	1457.1 ± 814.3	0.913

n—Frequency, %—Percentage, ±—Standard Deviation, €—Euro.

**Table 2 healthcare-13-03201-t002:** Spatiotemporal Gait Parameters and Their Association with Frailty Syndrome.

	Measure	Total (n = 99)	Not Frail (n = 39)	Prefrail (n = 51)	Frail (n = 9)	*p*-Value
**USUAL PACE**	Gait velocity (m/s) mean ±	1.04 ± 0.18	1.18 ± 0.11	0.96 ± 0.16	0.85 ± 0.11	<0.001
	Stride length (cm) mean ±	114.9 ± 15.9	126.7 ± 10.8	108.7 ± 13.3	98.5 ± 14.5	<0.001
	CV	4.9	4.2	5.1	6.1	0.058
	Stride time (s) mean ±	1.12 ± 0.10	1.07 ± 0.07	1.14 ± 0.10	1.17 ± 0.13	0.004
	CV	3.8	2.4	4.2	8.2	<0.001
	Support time (%)					
	mean ±	65.4 ± 2.6	63.9 ± 1.6	66.1 ± 2.6	68.1 ± 3.1	<0.001
	CV	3.0	2.1	3.4	5.1	<0.001
	Swing time (%)					
	mean ±	34.5 ± 2.5	36.0 ± 1.5	33.8 ± 2.4	31.9 ± 3.0	<0.001
	CV	6.3	3.7	7.3	11.7	<0.001
	Double support time (%) mean ±	30.4 ± 4.0	27.9 ± 3.1	31.7 ± 3.7	34.2 ± 3.2	<0.001
	CV	10.4	9.9	10.5	12.1	0.085
**FAST PACE**	Gait velocity (m/s) mean ±	1.31 ± 0.22	1.45 ± 0.14	1.24 ± 0.21	1.08 ± 0.18	<0.001
	Stride length (cm) mean ±	128.4 ± 18.1	139.7 ± 13.3	123.1 ± 16.4	109.3 ± 16.7	<0.001
	CV	4.6	4.5	4.6	5.6	0.246
	Stride time (s)					
	mean ±	0.98 ± 0.08	0.96 ± 0.07	1.00 ± 0.07	1.02 ± 0.11	0.019
	CV	3.1	2.7	3.1	4.3	<0.001
	Support time (%)					
	mean ±	63.3 ± 2.1	62.2 ± 1.6	63.8 ± 2.1	65.2 ± 2.1	<0.001
	CV	2.4	2.1	2.6	2.7	0.002
	Swing time (%) mean ±	36.6 ± 2.1	37.7 ± 1.6	36.1 ± 2.1	34.7 ± 2.1	<0.001
	CV	4.2	3.5	4.6	5.2	<0.001
	Double support time (%) mean ±	26.6 ± 4.2	24.5 ± 3.2	27.5 ± 4.1	30.4 ± 4.3	<0.000
	CV	11.6	11.2	12.0	10.3	<0.872

m/s—Meters/Second, ±—Standard Deviation, cm—Centimeters, s—Seconds, %—Percent age, CV—Coefficient of Varia.

**Table 3 healthcare-13-03201-t003:** Cognitive Function and Its Association with Frailty Syndrome.

	Total	Not Frail	Prefrail	Frail	*p*-Value
Measure	(n = 99)	(n = 39)	(n = 51)	(n = 9)	
MoCA mean ± SD	20.48 ± 3.99	21.85 ± 2.84	19.75 ± 4.35	18.78 ± 4.73	0.026
TMT-A mean ± SD	82.92 ± 50.95	70.52 ± 25.61	91.67 ± 64.28	87.11 ± 40.03	0.415
TMT-B mean ± SD	197.64 ± 86.08	177.56 ± 81.73	204.43 ± 89.63	246.11 ± 62.63	0.055
Stroop (interference) mean ± SD	−4.10 ± 6.59	−2.93 ± 7.10	−4.42 ± 6.03	−7.36 ± 6.71	0.208
DSB mean ± SD	3.62 ± 1.03	3.62 ± 0.99	3.75 ± 1.07	2.89 ± 0.78	0.071
DSC mean ± SD	16.30± 7.40	18.21 ± 7.19	15.51 ± 7.60	12.56 ± 5.15	0.090
DSF mean ± SD	5.17 ± 1.01	5.08 ± 0.87	5.20 ± 1.11	5.44 ± 1.01	0.589
TAVEC (n = 49)					
mean ± SD	7.08 ± 3.74	7.48 ± 3.88	7.05 ± 3.65	5.00 ± 4.08	0.508

MoCA—Montreal Cognitive Assessment, ±—Standard Deviation, TMT-A—Trail Making Test-A, TMT-B—Trail Making Test B, DSB—Digit Span Backward, DSC—Digit Symbol Coding, DSF—Digit Span Forward, TAVEC—Test Aprendizaje Verbal España Complutense: Long Term Free Recall Test.

**Table 4 healthcare-13-03201-t004:** Multivariate Regression Analysis Across Frailty Stages.

**Model 1: non-frail vs. prefrail**	**Measure**	**Coefficients B**	**OR (95% CI)**	***p*-Value**
	Sex (female)	1.503	4.94 (1.20–16.77)	0.025
	MCI (MoCA)	1.311	3.71 (1.08–12.69)	0.037
	Depressive symptoms (GDS-15)	0.600	1.82 (1.26–2.62)	0.001
	Mean stride length (fast pace)	−0.082	0.922 (0.88–0.96)	<0.001
**Model 2: prefrail vs. frail**	**MCI**	**MoCA**	**GDS-15**	**DSB**
		1.077	2.94 (1.34–6.44)	0.007
	DSB	−0.912	0.40 (0.16–0.97)	0.050

MCI—Mild Cognitive Impairment, MoCA—Montreal Cognitive Assessment, GDS-15—Geriatric Depression Scale Yesavage 15 Spanish version, DSB—Digit Span Backward.

## Data Availability

The dataset used is restricted and stored at the hospital where the study was conducted and can be obtained from the author upon reasonable request.

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
