# Peer review of "Linking Motor and Cognitive Decline in Aging: Gait Variability and Working Memory as Early Markers of Frailty"

_healthcare, 2025, doi:10.3390/healthcare13243201_

Round 1

Reviewer 1 Report

Comments and Suggestions for Authors

This study aimed to examine the relationship between frailty status, spatiotemporal gait parameters, and cognitive functions in community-dwelling older adults. However, the quality of the work is unsatisfactory as reflected by the following issues.

  1. Introduction fails to give previous studies on the prediction of early markers of frailty.
  2. What are the research questions?
  3. Why was the age of 70 selected for this study?
  4. The assessment of frailty was not clear.
  5. The details for the global cognitive function were missing.
  6. What are the independent and dependent variables for the ANOVA?
  7. The clinical practice was not closely related to the findings.
  8. The conclusion should be strengthened to highlight the main findings and the significance of the work.

Reviewer 2 Report

Comments and Suggestions for Authors

Frailty is a topic of continuing interest for those engaged in care of older adults, particularly in terms of providing the best care to help them maintain as much function as possible for as long as possible. On reading this paper, however, it becomes clear that there is considerable confusion regarding the concept of frailty, what it includes and what is does not include, whether it is considered a syndrome or another health-related phenomenon, if it has well identified underlying pathology, and so forth. For example, some factors such as vascular pathology could be causative while others could be the result of frailty.  More attention needs to be paid to the differences between these various associations with frailty. If these issues were addressed, the paper would be a much more substantial contribution to the literature than it is now.

A few additional points to consider in revising this paper:

More detail is needed on how the effect size was calculated.  In particular, I do not see the basis upon which it was calculated.  Did you use published data?  or your own data from prior studies? If based upon an assumption, what is the source for that assumption?

Caution is needed in talking about increases and decreases in symptoms, as if this were a longitudinal study and not a cross-sectional study.

Line 297: another caution, in this case against combining or confusing working memory with executive function and attention and putting too much emphasis on using subscales of the MoCA.

Readers might appreciate more information about preventing functional decline (line 350).

Round 2

Reviewer 1 Report

Comments and Suggestions for Authors

The authors have addressed my comments properly. 

Reviewer 2 Report

Comments and Suggestions for Authors

The authors have been responsive to reviewer comments. No further revision is needed.